# Scaling Marginalized Importance Sampling to High-Dimensional State-Spaces via State Abstraction

**Brahma S. Pavse and Josiah P. Hanna**
Department of Computer Science
University of Wisconsin – Madison, Madison, WI
pavse@wisc.edu, jphanna@cs.wisc.edu

## Abstract

We consider the problem of off-policy evaluation (OPE) in reinforcement learning (RL), where the goal is to estimate the performance of an evaluation policy, $\pi_e$, using a fixed dataset, $\mathcal{D}$, collected by one or more policies that may be different from $\pi_e$. Current OPE algorithms may produce poor OPE estimates under policy distribution shift i.e., when the probability of a particular state-action pair occurring under $\pi_e$ is very different from the probability of that same pair occurring in $\mathcal{D}$ Voloshin et al. (2021); Fu et al. (2021). In this work, we propose to improve the accuracy of OPE estimation by projecting the ground state-space into a lower-dimensional state-space using concepts from the state abstraction literature in RL. Specifically, we consider marginalized importance sampling (MIS) OPE algorithms which compute distribution correction ratios to produce their OPE estimate. In the original state-space, these ratios may have high variance which may lead to high variance OPE. However, we prove that in the lower-dimensional abstract state-space the ratios can have lower variance resulting in lower variance OPE. We then present a minimax optimization problem that incorporates the state abstraction. Finally, our empirical evaluation on difficult, high-dimensional state-space OPE tasks shows that the abstract ratios can make MIS OPE estimators achieve lower mean-squared error and more robust to hyperparameter tuning than the ground ratios.[1]

## 1 Introduction

This study focuses on the problem of off-policy evaluation (OPE) Fu et al. (2021); Voloshin et al. (2021), where the goal is to evaluate a policy of interest by leveraging offline data generated by possibly different policies. Solving the OPE problem would enable us to estimate the performance of a potentially risky policy without having to actually deploy it.

The core OPE problem is to produce accurate policy value estimates under policy distribution shift. This problem is particularly difficult on tasks with high-dimensional state-spaces Voloshin et al. (2021); Fu et al. (2021). For example, consider the AntUMaze problem illustrated on the left side of Figure 1. In this task, an ant-like robot with a high-dimensional state representation moves in a U-shaped maze and receives a reward only for reaching a specific 2D coordinate goal location. The state-space of this task includes information such as 2D location, ant limb angles, torso orientation etc., resulting in a 29-dimensional state-space. The OPE task is to evaluate the performance of a particular policy's ability to take the ant to the 2D goal location using data that may be collected by different policies. Policy distribution shift is common in this type of high-dimensional task since the chances of different policies inducing similar limb angles, torso orientations, paths traversed

---

[1] A fuller version of this paper will also be published at the Association for the Advancement of Artificial Intelligence (AAAI) 2023.

Offline Reinforcement Learning Workshop at Neural Information Processing Systems, 2022.

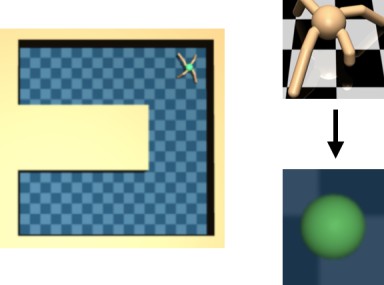

**Figure 1**: Left side: AntUMaze domain. Right side: Projecting high-dimensional ant into lower-dimensional point-mass.

etc. are incredibly slim. Notice, however, that while different policies may induce different body configurations, they may traverse similar 2D paths since all (successful) policies must move the ant through roughly the same path to reach the goal. Moreover, the only critical information needed from the state-space to determine the ant's per-step reward are its 2D coordinates. Motivated by this observation, we propose to improve the accuracy of OPE algorithms on high-dimensional state-space tasks by projecting the high-dimensional state-space into a lower-dimensional space. This idea is illustrated on the right side of Figure 1 where the ant is reduced to a 2D point-mass.

With this general motivation in mind, in this paper, we leverage concepts from the state abstraction literature Li et al. (2006) to improve the accuracy of marginalized importance sampling (MIS) OPE algorithms which estimate state-action density correction ratios to compute a policy value estimate Liu et al. (2018a); Xie et al. (2019). Due to the low chances of similarity between states of policies in high-dimensional state-spaces, current MIS algorithms can produce high variance state-action density ratios, resulting in high variance OPE estimates. However, if we are given a suitable state abstraction function, we can project the high-dimensional *ground* state-space into a lower-dimensional *abstract* state-space. The projection step increases the chances of similarity between these lower-dimensional states, resulting in low variance density ratios and OPE estimates.

## 2 Preliminaries

In this section, we discuss background information.

### 2.1 Notation and Problem Setup

We consider an infinite-horizon Markov decision process (MDP), $\mathcal{M} = \langle \mathcal{S}, \mathcal{A}, \mathcal{R}, P, \gamma, d_0 \rangle$, where $\mathcal{S}$ is the state-space, $\mathcal{A}$ is the action-space, $\mathcal{R} : \mathcal{S} \times \mathcal{A} \to \Delta([0, \infty))$ is the reward function, $P : \mathcal{S} \times \mathcal{A} \to \Delta(\mathcal{S})$ is the transition dynamics function, $\gamma \in [0, 1)$ is the discount factor, and $d_0 \in \Delta(\mathcal{S})$ is the initial state distribution. The agent acting, according to policy $\pi$, in the MDP generates a trajectory: $s_0, a_0, r_0, s_1, a_1, r_1, ...$, where $s_0 \sim d_0$, $a_t \sim \pi(\cdot|s_t)$, $r_t \sim \mathcal{R}(s_t, a_t)$, and $s_{t+1} \sim P(\cdot|s_t, a_t)$ for $t \geq 0$. We define $r(s, a) := \mathbb{E}_{r \sim \mathcal{R}(s,a)}[r]$ and the agent's discounted state-action occupancy measure under policy $\pi$ as $d_\pi(s, a) := \lim_{T \to \infty} \left( \sum_{t=0}^{T-1} \gamma^t d_\pi(s_t, a_t) \right) / \left( \sum_{t=0}^{T-1} \gamma^t \right)$, where $d_\pi(s_t, a_t)$ is the probability the agent will be in state $s$ and take action $a$ at time-step $t$ under policy $\pi$. Finally, we define the performance of policy $\pi$ to be its average reward, $\rho(\pi) := \mathbb{E}_{(s,a) \sim d_\pi, r \sim \mathcal{R}(s,a)}[r]$.

### 2.2 Off-Policy Evaluation (OPE)

In behavior-agnostic OPE, the goal is to estimate the performance of an evaluation policy $\pi_e$ given only a fixed offline data set of transition tuples, $\mathcal{D} := \{(s_i, a_i, s_i', r_i)\}_{i=1}^{mT}$, where $(s_i, a_i) \sim d_{\mathcal{D}}$, $m$ is the batch size (# of trajectories), and $T$ is the fixed length of each trajectory, generated by *unknown* and possibly *multiple* behavior policies. The difficulty in OPE is to estimate $\rho(\pi_e)$ under $d_{\pi_e}$ given samples only from $d_{\mathcal{D}}$.

We define the average-reward in dataset $\mathcal{D}$ to be $\bar{r}_{\mathcal{D}} := \mathbb{E}_{(s,a)\sim d_{\mathcal{D}}, r\sim\mathcal{R}(s,a)}[r]$. As in prior OPE work, we assume that if $d_{\pi_e}(s,a) > 0$ then $d_{\mathcal{D}}(s,a) > 0$. Empirically, we measure the accuracy of an estimate $\hat{\rho}(\pi_e)$ by generating $M$ datasets and then computing the *relative* mean-squared error (MSE): $\text{MSE}(\hat{\rho}(\pi_e)) := \frac{1}{M}\sum_{i=1}^{M} \frac{(\rho(\pi_e)-\hat{\rho}_i(\pi_e))^2}{(\rho(\pi_e)-\bar{r}_{\mathcal{D}_i})^2}$, where $\hat{\rho}_i(\pi_e)$ is computed using dataset $\mathcal{D}_i$ and $\bar{r}_{\mathcal{D}_i}$ is the average reward in $\mathcal{D}_i$.

### 2.2.1 Marginalized Importance Sampling (MIS)

In this work, we focus on MIS methods, which evaluate $\pi_e$ by using the ratio between $d_{\pi_e}$ and $d_{\mathcal{D}}$. That is, MIS methods evaluate $\pi_e$ by estimating $\rho(\pi_e) := \mathbb{E}_{(s,a)\sim d_{\mathcal{D}}, r\sim\mathcal{R}(s,a)}[\zeta(s,a)r]$, where $\zeta(s,a) := d_{\pi_e}(s,a)/d_{\mathcal{D}}(s,a)$ is the state-action density ratio for state-action pair $(s,a)$ and $d_{\pi}(s,a) = d_{\pi}(s)\pi(a|s)$. When the true $\zeta$ is known, the empirical estimate of $\rho(\pi_e)$ is:

$$\hat{\rho}(\pi_e) := \frac{1}{N}\sum_{i=1}^{N} \zeta(s_i, a_i) r(s_i, a_i) \tag{1}$$

where $N$ is the number of samples. In practice, however, $\zeta$ is unknown and must be estimated.

One set of $\zeta$-estimation algorithms is the DICE family Yang et al. (2020). While there are many variations, the general DICE optimization problem is:

$$\max_{\zeta:\mathcal{S}\times\mathcal{A}\to\mathbb{R}} \min_{\nu:\mathcal{S}\times\mathcal{A}\to\mathbb{R}} J(\zeta,\nu) :=$$
$$\mathbb{E}_{(s,a,s')\sim d_{\mathcal{D}}, a'\sim\pi_e}[\zeta(s,a)(\nu(s,a) - \gamma\nu(s',a'))] \tag{2}$$
$$- (1-\gamma)\mathbb{E}_{s_0\sim d_0, a_0\sim\pi_e}[\nu(s_0, a_0)]$$

where the solution to the optimization, $\zeta^*(s,a)$, are the true ratios. The estimator we present in Section 3 builds upon the DICE framework.

### 2.3 State Abstractions

We define a state abstraction function as a mapping $\phi : \mathcal{S} \to \mathcal{S}^\phi$, where $\mathcal{S}$ is called the *ground* state-space and $\mathcal{S}^\phi$ is called the *abstract* state-space. We consider state abstraction functions that partition the ground state-space into disjoint sets. We can use $\phi$ to project the original MDP into a new abstract MDP with the same action-space $\mathcal{A}$ and reward and transition dynamics functions defined as: $\mathcal{R}^\phi(s^\phi, a) = \sum_{s\in\phi^{-1}(s^\phi)} w(s)\mathcal{R}(s,a)$ and $P^\phi(s'^\phi|s^\phi, a) = \sum_{s\in\phi^{-1}(s^\phi), s'\in\phi^{-1}(s'^\phi)} w(s)P(s'|s,a)$, where $w : \mathcal{S} \to [0,1]$ is a ground state weighting function where for each $s^\phi$, $\sum_{s\in\phi^{-1}(s^\phi)} w(s) = 1$ Li et al. (2006). Similarly a policy can be transformed into its abstract equivalent as: $\pi^\phi(a|s^\phi) = \sum_{s\in\phi^{-1}(s^\phi)} w(s)\pi(a|s)$. In this work, we use $w_\pi(s) = \frac{d_\pi(s)}{\sum_{s'\in\phi^{-1}(s^\phi)} d_\pi(s')}$ and only consider abstractions that satisfy:

**Assumption 1** (Reward distribution equality). *$\forall s_1, s_2 \in \mathcal{S}$ such that $s_1, s_2 \in s^\phi$, $\forall a, \mathcal{R}(s_1, a) = \mathcal{R}(s_2, a)$.*

## 3 Abstract MIS

Marginalized IS methods may suffer from high variance in high-dimensional state-spaces. To potentially reduce this high variance, we propose to first use $\phi$ to project $\mathcal{D}$ into the abstract state-space to obtain: $\mathcal{D}^\phi := \{(s^\phi, a, r^\phi, s'^\phi)\}$ where $s^\phi = \phi(s)$ and $r^\phi(s,a) = r(s,a)\forall s \in s^\phi$, and then use the following estimator on $\mathcal{D}^\phi$ to estimate $\pi_e^\phi$:

$$\hat{\rho}(\pi_e^\phi) := \frac{1}{N}\sum_{i=1}^{N} \frac{d_{\pi_e^\phi}(s_i^\phi, a_i)}{d_{\mathcal{D}^\phi}(s_i^\phi, a_i)} r^\phi(s_i^\phi, a_i) \tag{3}$$

where $N$ is the number of samples, $d_{\pi^\phi}(s^\phi, a) = d_{\pi^\phi}(s^\phi)\pi^\phi(a|s^\phi)$ with $d_{\pi^\phi}(s^\phi) = \sum_{s\in\phi^{-1}(s^\phi)} d_\pi(s)$ and $\pi^\phi$ constructed using $w_\pi$.

In the remainder of this section, we present theoretical results on the statistical properties of the abstract ratios and the OPE estimator given in Equation (3). We then present a minimax optimization problem based on the DICE framework that incorporates state abstraction.

## 3.1 Theoretical Results

We now present the statistical properties of the abstract ratios and our estimator assuming it has access to the true abstract state-action ratios. Due to space constraints, we defer proofs to Appendix A.4.

We have Theorem 1, in which we prove that projecting $\mathcal{S} \to \mathcal{S}^\phi$ can lower the variance of density ratios:

**Theorem 1.** $Var\left(\frac{d_{\pi_e^\phi}(s^\phi, a)}{d_{\mathcal{D}^\phi}(s^\phi, a)}\right) \leq Var\left(\frac{d_{\pi_e}(s, a)}{d_{\mathcal{D}}(s, a)}\right)$

where equality holds only if $\phi$ is the identity function i.e. $\phi(s) = s, \forall s \in \mathcal{S}$ and/or if $\forall s_1, s_2 \in \mathcal{S}$ such that $\forall s_1, s_2 \in s^\phi$ and for a given action $a$, $\frac{d_{\pi_e}(s_1, a)}{d_{\mathcal{D}}(s_1, a)} = \frac{d_{\pi_e}(s_2, a)}{d_{\mathcal{D}}(s_2, a)}, \forall s^\phi \in \mathcal{S}^\phi, a \in \mathcal{A}$.

Furthermore, we prove our abstract estimator is unbiased (Theorem 3 in Appendix A.4) and strongly consistent (Theorem 2 and Corollary 1):

**Theorem 2.** Our estimator, $\hat{\rho}(\pi_e^\phi)$, given in Equation 3 is an asymptotically consistent estimator of $\rho(\pi_e)$ in terms of MSE: $\lim_{N \to \infty} \mathbb{E}[(\hat{\rho}(\pi_e^\phi) - \rho(\pi_e))^2] = 0$.

## 3.2 MIS OPE with Abstract DICE

To test the effectiveness of state abstraction for OPE, we evaluate the DICE framework on $\mathcal{D}^\phi$. We focus on BestDICE Yang et al. (2020), and call our algorithm AbstractBestDICE, which solves the following optimization problem:

$$
\begin{aligned}
\min_{\nu, \lambda} \max_\zeta J(\nu, \zeta, \lambda) :=& -\mathbb{E}_{\mathcal{D}^\phi}\left[\frac{1}{2}\zeta(s^\phi, a)^2\right] \\
&+ \mathbb{E}_{\mathcal{D}^\phi}\left[\zeta(s^\phi, a)\left(\gamma \mathbb{E}_{a' \sim \pi_e^\phi}[\nu(s'^\phi, a')] - \nu(s^\phi, a) - \lambda\right)\right] \\
&+ (1 - \gamma)\mathbb{E}_{s_0^\phi \sim d_0^\phi, a_0 \sim \pi_e^\phi}[\nu(s_0^\phi, a_0)] + \lambda
\end{aligned}
\tag{4}
$$

where $\nu : \mathcal{S}^\phi \times \mathcal{A} \to \mathbb{R}$, $\lambda \in \mathbb{R}$, and $\zeta : \mathcal{S}^\phi \times \mathcal{A} \to \mathbb{R}_{\geq 0}$. The solution to this optimization, $\zeta^*(s^\phi, a)$, is then plugged into the estimator given in Equation (3) to get an OPE estimate. Note that we have not proven whether this optimization recovers the true abstract density ratios. However, in Section 4 we show that AbstractBestDICE can still lead to accurate OPE in high-dimensional state-spaces.

# 4 Empirical Study

We will now show how projecting $\mathcal{S} \to \mathcal{S}^\phi$ can produce data-efficient and stable OPE estimates in practice.

## 4.1 Empirical Setup

In this section, we describe the algorithms and domains of our empirical study. Due to space constraints, we defer supporting details to the appendix (A.5 and A.6).

### 4.1.1 Algorithms

We compare AbstractBestDICE to ground BestDICE. As also reported by Yang et al. (2020); Fu et al. (2021), we found in preliminary experiments that BestDICE performed much better than the other DICE variants such as DualDICE Nachum et al. (2019), GenDICE Zhang et al. (2020a), etc.

### 4.1.2 Domains

We focus on high-dimensional state-space tasks, which have been known to be particularly challenging for DICE methods Fu et al. (2021). We specify the fixed $\phi$ for each environment in Appendix A.6.

- **Reacher**. A robotic arm tries to move to a goal location. Here, $s \in \mathbb{R}^{11}$, $a \in \mathbb{R}^2$, and $s^\phi \in \mathbb{R}^4$.

- **Walker2D**. A bi-pedal robot tries to move as fast as possible. Here, $s \in \mathbb{R}^{18}$, $a \in \mathbb{R}^6$, and $s^\phi \in \mathbb{R}^3$.

- **Pusher**. A robotic arm tries to push an object to a goal location. Here, $s \in \mathbb{R}^{23}$, $a \in \mathbb{R}^7$, and $s^\phi \in \mathbb{R}^6$.

- **AntUMaze**. This sparse-reward task requires an ant to move from one end of the U-shaped maze to the other end. Here, $s \in \mathbb{R}^{29}$, $a \in \mathbb{R}^8$, and $s^\phi \in \mathbb{R}^2$.

## 4.2 Empirical Results

In this section, we describe our main empirical results; additional experiments can be found in appendix A.6.

### 4.2.1 Data-Efficiency

Figure 2 shows the results of our (relative) MSE vs. batch size experiment for the function approximation case. For a given batch size, we train each algorithm for 100k epochs with different hyperparameters sets, record the (relative) MSE on the last epoch by each hyperparameter set, and plot the lowest MSE achieved by these hyperparameter sets. We find that AbstractBestDICE is able to achieve lower MSE than BestDICE for a given batch size. We note that while hyperparameter tuning is difficult in OPE, in this experiment, we aim to evaluate each algorithm assuming each had favorable hyperparameters.

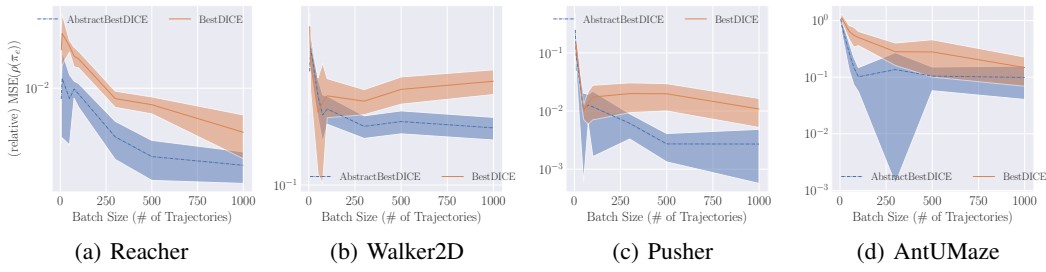

| (a) Reacher | (b) Walker2D | (c) Pusher | (d) AntUMaze |

**Figure 2**: Relative MSE vs. Batch Size (# of trajectories). Vertical axis is log-scaled. Errors are computed over 15 trials with 95% confidence intervals. Lower is better.

### 4.2.2 Hyperparameter Robustness

Finally, we study the robustness of these algorithms to hyperparameters tuning. In practical OPE, hyperparameter tuning with respect to MSE is impractical since the true $\rho(\pi_e)$ is unknown Fu et al. (2021); Paine et al. (2020). Thus, we want OPE algorithms to be as robust as possible to

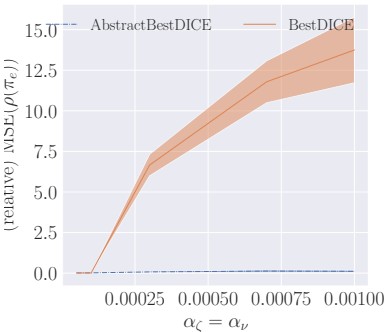

**Figure 3**: Robustness of BestDICE and AbstractBestDICE to hyperparameters on the Pusher domain for batch size (# of trajectories) of 50. Errors are computed over 15 trials with 95% confidence intervals. Lower is better.

hyperparameter tuning. The main hyperparameters for DICE are the learning rates of $\zeta$ and $\nu$, $\alpha_\zeta$ and $\alpha_\nu$. We focus on small batch sizes, where we would expect high sensitivity. The results of this study are in Figure 3. We find that AbstractBestDICE has a less volatile MSE than BestDICE (also see appendix A.6).

## 5  Related Work

**MIS and Off-Policy Evaluation**. There have been broadly three families of MIS algorithms in the OPE literature to estimate state-action density ratios. One is the DICE family, which includes: minimax-weight learning Uehara et al. (2019), DualDice Nachum et al. (2019), GenDICE Zhang et al. (2020a), GradientDICE Zhang et al. (2020b), and BestDICE Yang et al. (2020). The second family of MIS algorithms is the COP-TD algorithm Hallak and Mannor (2017); Gelada and Bellemare (2019), which learns the state density ratios with an online TD-styled update. The third family is the variational power method Wen et al. (2020) algorithm which generalizes the power iteration method to estimate density ratios. While our focus has been on MIS algorithms, there are many other OPE algorithms such as model-based methods Zhang et al. (2021b); Hanna et al. (2017); Liu et al. (2018b), fitted-Q evaluation Le et al. (2019), and IS Precup et al. (2000); Thomas (2015); Hanna et al. (2019).

**State Abstraction**. The literature on state abstraction is extensive Singh et al. (1994); Dietterich (1999); Ferns et al. (2011); Li et al. (2006); Abel (2020). However, much of this work has been exclusively focused on building a theory of abstraction and learning optimal policies. To the best of our knowledge, no work has leveraged state abstraction to improve the accuracy of OPE algorithms.

## 6  Summary and Future Work

In this work, we showed that we can improve the accuracy of OPE estimates by projecting the original ground state-space into a lower-dimensional abstract state-space using state abstraction and performing OPE in the resulting abstract Markov decision process. We showed that AbstractBestDICE obtained more accurate estimates with added hyperparameter robustness on difficult, high-dimensional state-space tasks.

As for future work, it would be interesting to leverage existing ideas Gelada et al. (2019); Zhang et al. (2021a) to learn $\phi$ instead of using a fixed $\phi$. Another interesting direction would be to apply abstraction to other OPE algorithms. While this work focused exclusively on MIS algorithms, a promising direction will be to apply abstraction techniques to model-based, trajectory IS, and value-function based OPE.

## Acknowledgements

Support for this research was provided by the Office of the Vice Chancellor for Research and Graduate Education at the University of Wisconsin — Madison with funding from the Wisconsin Alumni Research Foundation. The authors thank the anonymous reviewers, Nicholas Corrado, Ishan Durugkar, and Subhojyoti Mukherjee for their helpful comments in improving this work.

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

# A Appendix

## A.1 A Hard Example for Ground MIS Ratios

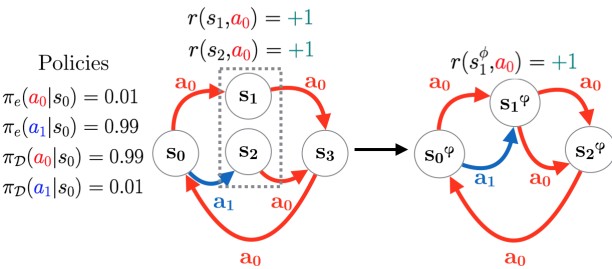

**Figure 4**: TwoPath MDP where ground density ratios for $(s_1, a_0)$ and $(s_2, a_0)$ are high variance. However, upon aggregation of equivalent states (grey dotted lines), the abstract density ratio of $(s_1^\phi, a_0)$ is low variance.

We present a hard example for ground MIS ratios in Figure 4 that provides intuition for why the abstract MIS ratios can have lower variance ratios than the ground ratios. Consider two symmetric policies, $\pi_e$ and $\pi_\mathcal{D}$, executed in the ground MDP (left side). In this example, the high variance of the true state-action density ratios $\frac{d_{\pi_e}(s_1, a_0)}{d_{\pi_\mathcal{D}}(s_1, a_0)} \approx 0$ and $\frac{d_{\pi_e}(s_2, a_0)}{d_{\pi_\mathcal{D}}(s_2, a_0)} \approx 100$ can lead to high variance estimates of $\rho(\pi_e)$. Notice, however, that states $s_1$ and $s_2$ are essentially equivalent i.e. $r(s_1, a) = r(s_2, a) \forall a \in \mathcal{A}$ and can be aggregated together into a single state, $s_1^\phi$ (Assumption 1). We find that the state-action density ratio in this abstract MDP (right side) $\frac{d_{\pi_e^\phi}(s_1^\phi, a_0)}{d_{\pi_\mathcal{D}^\phi}(s_1^\phi, a_0)} = 1$ is of low variance, which can lead to low variance estimates of $\rho(\pi_e)$.

## A.2 Preliminaries

This section provides the supporting lemmas and definitions that we leverage to prove our lemmas and theorems.

**Definition 1** (Almost Sure Convergence). *A sequence of random variables, $(X_n)_{n=1}^\infty$, almost surely converges to the random variable, $X$ if*

$$\mathbb{P}\left( \lim_{n \to \infty} X_n = X \right) = 1$$

*We write $X_n \overset{a.s.}{\to} X$ to denote that the sequence $(X_n)_{n=1}^\infty$ converges almost surely to $X$.*

**Definition 2** ((Strongly) Consistent Estimator). *Let $\theta$ be a real number and $(\hat{\theta}_n)_{n=1}^\infty$ be an infinite sequence of random variables. We call $\hat{\theta}_n$ a (strongly) consistent estimator of $\theta$ if and only if $\hat{\theta}_n \overset{a.s.}{\to} \theta$.*

**Lemma 1.** *If $(X_i)_{i=1}^\infty$ is a sequence of uniformly bounded real-valued random variables, then $X_n \overset{a.s.}{\to} X$ if and only if $\lim_{N \to \infty} \mathbb{E}[(X_n - X)^2] = 0$.*

**Proof** See Lemma 3 in Thomas and Brunskill (2016). $\qquad\square$

## A.3 Assumptions and Definitions

In the main paper, we provided the major assumptions required for our theoretical and empirical work relevant to abstraction and OPE. Here we provide supporting assumptions typically used in the OPE literature used for the theoretical analysis.

**Assumption 2** (Coverage). *For all $(s, a) \in \mathcal{S} \times \mathcal{A}$, if $\pi_e(a|s) > 0$ then $\pi_b(a|s) > 0$.*

**Assumption 3** (Non-negative reward). *We assume that the reward function is bounded between $[0, \infty)$.*

**Definition 3** (Ground state normalized weightings). *For a given policy $\pi$, each ground state $s \in s^\phi$, has a state aggregation weight, $w_\pi(s) = \frac{d_\pi(s)}{\sum_{s' \in \phi^{-1}(s^\phi)} d_\pi(s')}$, where $d_\pi(s)$ is the discounted state-occupancy measure of $\pi$.*

## A.4 Proofs

In the proofs below, we denote the collection of behavior policies that generated $\mathcal{D}$ with $\pi_\mathcal{D}$. That is, $\pi_\mathcal{D}$ is the conditional probability of an action occurring in a given state in the data. Similarly, we also have $\pi_\mathcal{D}^\phi$. These minor changes give us $d_\mathcal{D} = d_{\pi_\mathcal{D}}$ and $d_{\mathcal{D}^\phi} = d_{\pi_\mathcal{D}^\phi}$.

**Lemma 2.** *For an arbitrary function, $f$, $\mathbb{E}_{s^\phi \sim d_{\pi^\phi}, a \sim \pi^\phi}\left[f(s^\phi, a)\right] = \mathbb{E}_{s \sim d_\pi, a \sim \pi}\left[f(\phi(s), a)\right]$.*

**Proof**

$$\mathbb{E}_{s^\phi \sim d_{\pi^\phi}, a \sim \pi^\phi}\left[f(s^\phi, a)\right] = \sum_{s^\phi, a} d_{\pi^\phi}(s^\phi) \pi^\phi(a|s^\phi) f(s^\phi, a)$$

$$\overset{(a)}{=} \sum_{s^\phi, a} d_{\pi^\phi}(s^\phi) \sum_{s \in \phi^{-1}(s^\phi)} \frac{\pi(a|s) d_\pi(s)}{d_{\pi^\phi}(s^\phi)} f(s^\phi, a)$$

$$= \sum_{s^\phi, a} \sum_{s \in \phi^{-1}(s^\phi)} \pi(a|s) d_\pi(s) f(s^\phi, a)$$

$$= \sum_{s, a} \pi(a|s) d_\pi(s) f(\phi(s), a)$$

$$\mathbb{E}_{s^\phi \sim d_\pi^\phi, a \sim \pi^\phi}\left[f(s^\phi, a)\right] = \mathbb{E}_{s \sim d_\pi, a \sim \pi}\left[f(\phi(s), a)\right]$$

where (a) is due to Definition 3 and we can replace $s^\phi$ with $\phi(s)$ when we know $s \in s^\phi$. $\qquad\square$

**Theorem 1.** *$Var\left(\frac{d_{\pi_e^\phi}(s^\phi, a)}{d_{\mathcal{D}^\phi}(s^\phi, a)}\right) \le Var\left(\frac{d_{\pi_e}(s, a)}{d_\mathcal{D}(s, a)}\right)$*

**Proof**

Before comparing the variances, we note that due to Assumption 2 and Lemma 2:

$$\mathbb{E}_{s \sim d_{\pi_\mathcal{D}}, a \sim \pi_\mathcal{D}}\left[\frac{d_{\pi_e}(s) \pi_e(a|s)}{d_{\pi_\mathcal{D}}(s) \pi_\mathcal{D}(a|s)}\right] = \mathbb{E}_{s \sim d_{\pi_\mathcal{D}^\phi}, a \sim \pi_\mathcal{D}^\phi}\left[\frac{d_{\pi_e^\phi}(s^\phi) \pi_e^\phi(a|s^\phi)}{d_{\pi_\mathcal{D}^\phi}(s^\phi) \pi_\mathcal{D}^\phi(a|s^\phi)}\right] = 1$$

Denote, $V^g := Var\left(\frac{d_{\pi_e}(s) \pi_e(a|s)}{d_{\pi_\mathcal{D}}(s) \pi_\mathcal{D}(a|s)}\right)$ and $V^\phi := Var\left(\frac{d_{\pi_e^\phi}(s^\phi) \pi_e^\phi(a|s^\phi)}{d_{\pi_\mathcal{D}^\phi}(s^\phi) \pi_\mathcal{D}^\phi(a|s^\phi)}\right)$. Now consider the difference between the two variances.

$D = V^g - V^\phi$

$$= Var\left(\frac{d_{\pi_e}(s) \pi_e(a|s)}{d_{\pi_\mathcal{D}}(s) \pi_\mathcal{D}(a|s)}\right) - Var\left(\frac{d_{\pi_e^\phi}(s^\phi) \pi_e^\phi(a|s^\phi)}{d_{\pi_\mathcal{D}^\phi}(s^\phi) \pi_\mathcal{D}^\phi(a|s^\phi)}\right)$$

$$= \mathbb{E}_{s \sim d_{\pi_\mathcal{D}}, a \sim \pi_\mathcal{D}}\left[\left(\frac{d_{\pi_e}(s) \pi_e(a|s)}{d_{\pi_\mathcal{D}}(s) \pi_\mathcal{D}(a|s)}\right)^2\right] - \mathbb{E}_{s^\phi \sim d_{\pi_\mathcal{D}^\phi}, a \sim \pi_\mathcal{D}^\phi}\left[\left(\frac{d_{\pi_e^\phi}(s^\phi) \pi_e^\phi(a|s^\phi)}{d_{\pi_\mathcal{D}^\phi}(s^\phi) \pi_\mathcal{D}^\phi(a|s^\phi)}\right)^2\right]$$

$$= \sum_{s, a} d_{\pi_\mathcal{D}}(s) \pi_\mathcal{D}(a|s) \left(\frac{d_{\pi_e}(s) \pi_e(a|s)}{d_{\pi_\mathcal{D}}(s) \pi_\mathcal{D}(a|s)}\right)^2 - \sum_{s^\phi, a} d_{\pi_\mathcal{D}^\phi}(s^\phi) \pi_\mathcal{D}^\phi(a|s^\phi) \left(\frac{d_{\pi_e^\phi}(s^\phi) \pi_e^\phi(a|s^\phi)}{d_{\pi_\mathcal{D}^\phi}(s^\phi) \pi_\mathcal{D}^\phi(a|s^\phi)}\right)^2$$

$$= \sum_{s^\phi, a} \left(\sum_{s \in s^\phi} d_{\pi_\mathcal{D}}(s) \pi_\mathcal{D}(a|s) \left(\frac{d_{\pi_e}(s) \pi_e(a|s)}{d_{\pi_\mathcal{D}}(s) \pi_\mathcal{D}(a|s)}\right)^2 - d_{\pi_\mathcal{D}^\phi}(s^\phi) \pi_\mathcal{D}^\phi(a|s^\phi) \left(\frac{d_{\pi_e^\phi}(s^\phi) \pi_e^\phi(a|s^\phi)}{d_{\pi_\mathcal{D}^\phi}(s^\phi) \pi_\mathcal{D}^\phi(a|s^\phi)}\right)^2\right)$$

We can analyze this difference by looking at one abstract state and one action and all the states that belong to it. That is, for a fixed abstract state, $s^\phi$, and fixed action, $a$, we have:

$$D' = \sum_{s \in s^\phi} d_{\pi_\mathcal{D}}(s) \pi_\mathcal{D}(a|s) \left( \frac{d_{\pi_e}(s)\pi_e(a|s)}{d_{\pi_\mathcal{D}}(s)\pi_\mathcal{D}(a|s)} \right)^2 - \left( d_{\pi_\mathcal{D}^\phi}(s^\phi)\pi_\mathcal{D}^\phi(a|s^\phi) \left( \frac{d_{\pi_e^\phi}(s^\phi)\pi_e^\phi(a|s^\phi)}{d_{\pi_\mathcal{D}^\phi}(s^\phi)\pi_\mathcal{D}^\phi(a|s^\phi)} \right)^2 \right)$$

$$= \left( \sum_{s \in s^\phi} \frac{(d_{\pi_e}(s)\pi_e(a|s))^2}{d_{\pi_\mathcal{D}}(s)\pi_\mathcal{D}(a|s)} \right) - \left( \frac{(d_{\pi_e^\phi}(s^\phi)\pi_e^\phi(a|s^\phi))^2}{d_{\pi_\mathcal{D}^\phi}(s^\phi)\pi_\mathcal{D}^\phi(a|s^\phi)} \right)$$

$$\overset{(a)}{=} \left( \left( \sum_{s \in s^\phi} \frac{(d_{\pi_e}(s)\pi_e(a|s))^2}{d_{\pi_\mathcal{D}}(s)\pi_\mathcal{D}(a|s)} \right) - \left( \frac{(d_{\pi_e^\phi}(s^\phi)\pi_e^\phi(a|s^\phi)^2}{d_{\pi_\mathcal{D}^\phi}(s^\phi)\pi_\mathcal{D}^\phi(a|s^\phi)} \right) \right)$$

where (a) is due to Definition 3.

If we can show that $D' \geq 0$ for all possible sizes of $|s^\phi|$, we will the have the original difference, $D$, is a sum of only non-negative terms, thus proving Theorem 1. We will prove $D' \geq 0$ by inductive proof on the size of $|s^\phi|$ from 1 to some $n \leq |\mathcal{S}|$.

Let our statement to prove, $P(n)$ be that $D' \geq 0$ where $n = |s^\phi|$. This is trivially true for $P(1)$ where the ground state equals the abstract state. Now consider the inductive hypothesis, $P(n)$ is true for $n \geq 1$. Now with the inductive step, we must show that $P(n+1)$ is true given $P(n)$ is true. Starting with the inductive hypothesis:

$$D'' = \underbrace{\left( \sum_{s \in s^\phi} \frac{(d_{\pi_e}(s)\pi_e(a|s))^2}{d_{\pi_\mathcal{D}}(s)\pi_\mathcal{D}(a|s)} \right)}_{S} - \left( \frac{\overbrace{((d_{\pi_e^\phi}(s^\phi)\pi_e^\phi(a|s^\phi))^2}^{C}}{\underbrace{d_{\pi_\mathcal{D}^\phi}(s^\phi)\pi_\mathcal{D}^\phi(a|s^\phi)}_{C'}} \right) \geq 0$$

We define $S := \left( \sum_{s \in s^\phi} \frac{(d_{\pi_e}(s)\pi_e(a|s))^2}{d_{\pi_\mathcal{D}}(s)\pi_\mathcal{D}(a|s)} \right)$, $C := (d_{\pi_e^\phi}(s^\phi)\pi_e^\phi(a|s^\phi)$, and $C' := d_{\pi_\mathcal{D}^\phi}(s^\phi)\pi_\mathcal{D}^\phi(a|s^\phi)$. After making the substitutions, we have:

$$C^2 \leq SC' \tag{5}$$

We have the above result holding true for when the $|s^\phi| = n$. Now consider the inductive step in relation to the inductive hypothesis where a new state, $s_{n+1}$ is added to the abstract state. We have the following difference:

$$D'' = S + \frac{(d_{\pi_e}(s_{n+1})\pi_e(a|s_{n+1}))^2}{d_{\pi_\mathcal{D}}(s_{n+1})\pi_\mathcal{D}(a|s_{n+1})} - \frac{(C + d_{\pi_e}(s_{n+1})\pi_e(a|s_{n+1}))^2}{C' + d_{\pi_\mathcal{D}}(s_{n+1})\pi_\mathcal{D}(a|s_{n+1})}$$

For ease in notation, let $x = d_{\pi_e}(s_{n+1})\pi_e(a|s_{n+1})$ and $y = d_{\pi_{\mathcal{D}}}(s_{n+1})\pi_{\mathcal{D}}(a|s_{n+1})$. The above difference is then:

$$
\begin{aligned}
D'' &= S + \frac{x^2}{y} - \frac{(C+x)^2}{C'+y} \\
&= \frac{Sy + x^2}{y} - \frac{(C+x)^2}{C'+y} \\
&= \frac{1}{y(C'+y)}((Sy + x^2)(C'+y) - (C+x)^2 y) \\
&= \frac{1}{y(C'+y)}(SyC' + Sy^2 + x^2 C' + x^2 y - C^2 y - x^2 y - 2Cxy) \\
&= \frac{1}{y(C'+y)}(SyC' + Sy^2 + x^2 C' - C^2 y - 2Cxy)
\end{aligned}
$$

The above difference, $D''$, is minimized most when $C$ is as large as possible. From the inductive hypothesis, we have $C \le \sqrt{SC'}$. The minimum difference can be written as:

$$
\begin{aligned}
D'' &= \frac{1}{y(C'+y)}(Sy^2 + x^2 C' - 2\sqrt{SC'}xy) \\
&= \frac{1}{y(C'+y)}(y\sqrt{S} - x\sqrt{C'})^2 \\
&\ge 0
\end{aligned}
$$

So we have $D'' \ge 0$ for $|s^\phi| = n + 1$, which means $D' \ge 0$. We have showed that $P(n)$ is true for all $n$. We now have the original difference, $D$, to be a sum of non-negative terms after performing this same grouping for all abstract states and actions, which results in:

$$
\mathrm{Var}\left(\frac{d_{\pi_e^\phi}(s^\phi)\pi_e^\phi(a|s^\phi)}{d_{\pi_{\mathcal{D}}^\phi}(s^\phi)\pi_{\mathcal{D}}^\phi(a|s^\phi)}\right) \le \mathrm{Var}\left(\frac{d_{\pi_e}(s)\pi_e(a|s)}{d_{\pi_{\mathcal{D}}}(s)\pi_{\mathcal{D}}(a|s)}\right)
$$

Thus, we have:

$$
\mathrm{Var}\left(\frac{d_{\pi_e^\phi}(s^\phi, a)}{d_{\pi_{\mathcal{D}}^\phi}(s^\phi, a)}\right) \le \mathrm{Var}\left(\frac{d_{\pi_e}(s, a)}{d_{\pi_{\mathcal{D}}}(s, a)}\right)
$$

$\square$

**Proposition 1.** *If Assumption 1 holds, the average reward of ground policy $\pi$ executed in ground MDP $\mathcal{M}$, $\rho(\pi)$, is equal to the average reward of abstract policy $\pi^\phi$ executed in abstract MDP $\mathcal{M}^\phi$ constructed with $w_\pi$, $\rho(\pi^\phi)$. That is, $\rho(\pi) = \rho(\pi^\phi)$.*

**Proof** Consider the definition of $R_\pi^\phi$:

$$\rho(\pi^\phi) = \sum_{s^\phi,a} d_{\pi^\phi}(s^\phi)\pi^\phi(a|s^\phi)r(s^\phi,a)$$

$$= \sum_{s^\phi,a}\left(\left(\sum_{s\in\phi^{-1}(s^\phi)} d_\pi(s)\right)\left(\sum_{s\in\phi^{-1}(s^\phi)}\pi(a|s)w_\pi(s)\right)r(s^\phi,a)\right)$$

$$\overset{(a)}{=} \sum_{\phi(s),a}\left(\left(\sum_{s\in\phi^{-1}(s^\phi)} d_\pi(s)\right)\left(\sum_{s\in\phi^{-1}(s^\phi)}\frac{\pi(a|s)d_\pi(s)}{\sum_{s'\in\phi^{-1}(s^\phi)} d_\pi(s')}\right)r(s^\phi,a)\right)$$

$$\overset{(b)}{=} \sum_{\phi(s),a}\left(\sum_{s\in\phi^{-1}(s^\phi)}\pi(a|s)d_\pi(s)\right)r(s^\phi,a)$$

$$\overset{(c)}{=} \sum_{\phi(s),a}\left(\sum_{s\in\phi^{-1}(s^\phi)}\pi(a|s)d_\pi(s)r(s,a)\right)$$

$$= \sum_{s,a}\pi(a|s)d_\pi(s)r(s,a)$$

$$\rho(\pi^\phi) = \rho(\pi)$$

where (a) is due to Definition 3, (b) is due to Definition 3 and Assumption 1 $\qquad\square$

**Theorem 3.** *If Assumption 1 holds, our estimator, $\hat{\rho}(\pi_e^\phi)$ as defined in Equation 3, is an unbiased estimator of $\rho(\pi_e)$.*

**Proof**

We first consider the expectation of a single sample, $X = \frac{d_{\pi_e^\phi}(s^\phi)\pi_e^\phi(a|s^\phi)}{d_{\pi_\mathcal{D}^\phi}(s^\phi)\pi_\mathcal{D}^\phi(a|s^\phi)}r^\phi(s^\phi,a)$:

$$\mathbb{E}_{s\sim d_{\pi_\mathcal{D}},a\sim\pi_\mathcal{D}}[X] = \sum_{s,a} d_{\pi_\mathcal{D}}(s)\pi_\mathcal{D}(a|s)\frac{d_{\pi_e^\phi}(s^\phi)\pi_e^\phi(a|s^\phi)}{d_{\pi_\mathcal{D}^\phi}(s^\phi)\pi_\mathcal{D}^\phi(a|s^\phi)}r^\phi(s^\phi,a)$$

$$\overset{(a)}{=} \sum_{s^\phi,a}\sum_{s\in\phi^{-1}(s^\phi)} d_{\pi_\mathcal{D}}(s)\pi_\mathcal{D}(a|s)\frac{d_{\pi_e^\phi}(s^\phi)\pi_e^\phi(a|s^\phi)}{d_{\pi_\mathcal{D}^\phi}(s^\phi)\pi_\mathcal{D}^\phi(a|s^\phi)}r^\phi(s^\phi,a)$$

$$\overset{(b)}{=} \sum_{s^\phi,a}\frac{d_{\pi_e^\phi}(s^\phi)\pi_e^\phi(a|s^\phi)}{d_{\pi_\mathcal{D}^\phi}(s^\phi)\pi_\mathcal{D}^\phi(a|s^\phi)}\sum_{s\in\phi^{-1}(s^\phi)} d_{\pi_\mathcal{D}}(s)\pi_\mathcal{D}(a|s)r^\phi(s^\phi,a)$$

$$\overset{(c)}{=} \sum_{s^\phi,a}\frac{d_{\pi_e^\phi}(s^\phi)\pi_e^\phi(a|s^\phi)}{d_{\pi_\mathcal{D}^\phi}(s^\phi)\pi_\mathcal{D}^\phi(a|s^\phi)}r^\phi(s^\phi,a)\sum_{s\in\phi^{-1}(s^\phi)} d_{\pi_\mathcal{D}}(s)\pi_\mathcal{D}(a|s)$$

$$= \sum_{s^\phi,a}\frac{d_{\pi_e^\phi}(s^\phi)\pi_e^\phi(a|s^\phi)}{d_{\pi_\mathcal{D}^\phi}(s^\phi)\pi_\mathcal{D}^\phi(a|s^\phi)}r^\phi(s^\phi,a)\sum_{s'\in\phi^{-1}(s^\phi)} d_{\pi_\mathcal{D}}(s')\sum_{s\in\phi^{-1}(s^\phi)}\frac{d_{\pi_\mathcal{D}}(s)\pi_\mathcal{D}(a|s)}{\sum_{s'\in\phi^{-1}(s^\phi)} d_{\pi_\mathcal{D}}(s')}$$

$$= \sum_{s^\phi,a}\frac{d_{\pi_e^\phi}(s^\phi)\pi_e^\phi(a|s^\phi)}{d_{\pi_\mathcal{D}^\phi}(s^\phi)\pi_\mathcal{D}^\phi(a|s^\phi)}r^\phi(s^\phi,a)(d_{\pi_\mathcal{D}^\phi}(s^\phi)\pi_\mathcal{D}^\phi(a|s^\phi))$$

$$= \sum_{s^\phi,a} d_{\pi_e^\phi}(s^\phi)\pi_e^\phi(a|s^\phi)r^\phi(s^\phi,a)$$

$$\overset{(d)}{=} \rho(\pi_e^\phi)$$

$$\mathbb{E}_{s\sim d_{\pi_\mathcal{D}},a\sim\pi_\mathcal{D}}[X] \overset{(e)}{=} \rho(\pi_e)$$

where (c) is due to Definition 3 and Assumption 1, (d) is due to Assumption 2, and (e) is due to Proposition 1.

We have the bias defined as:

$$\text{Bias}[\hat{\rho}(\pi_e^\phi)] = \mathbb{E}_{s \sim d_{\pi_{\mathcal{D}}}, a \sim \pi_{\mathcal{D}}}[\hat{\rho}(\pi_e^\phi)] - R_{\pi_e}$$

$$= \mathbb{E}_{s \sim d_{\pi_{\mathcal{D}}}, a \sim \pi_{\mathcal{D}}} \left[ \frac{1}{mT} \sum_{i=1}^{mT} \frac{d_{\pi_e^\phi}(s_i^\phi) \pi_e^\phi(a_i|s_i^\phi)}{d_{\pi_{\mathcal{D}}^\phi}(s_i^\phi) \pi_{\mathcal{D}}^\phi(a_i|s_i^\phi)} r^\phi(s_i^\phi, a_i) \right] - R_{\pi_e}$$

$$\overset{(a)}{=} \frac{1}{mT} \sum_{i=1}^{mT} \mathbb{E}_{s_i \sim d_{\pi_{\mathcal{D}}}, a_i \sim \pi_{\mathcal{D}}} \left[ \frac{d_{\pi_e^\phi}(s_i^\phi) \pi_e^\phi(a_i|s_i^\phi)}{d_{\pi_{\mathcal{D}}^\phi}(s_i^\phi) \pi_{\mathcal{D}}^\phi(a_i|s_i^\phi)} r^\phi(s_i^\phi, a_i) \right] - R_{\pi_e}$$

$$\overset{(b)}{=} \left( \frac{1}{mT} \sum_{i=1}^{mT} R_{\pi_e} \right) - R_{\pi_e}$$

$$= \rho(\pi_e) - \rho(\pi_e)$$

$$\text{Bias}[\hat{\rho}(\pi_e^\phi)] = 0$$

where (a) is due to linearity of expectation and (b) is due to expectation of a single sample. □

**Theorem 2.** *Our estimator, $\hat{\rho}(\pi_e^\phi)$, given in Equation 3 is an asymptotically consistent estimator of $\rho(\pi_e)$ in terms of MSE:* $\lim_{N \to \infty} \mathbb{E}[(\hat{\rho}(\pi_e^\phi) - \rho(\pi_e))^2] = 0$.

**Proof** We have the MSE of $\hat{\rho}(\pi_e^\phi)$ w.r.t $\rho(\pi_e)$ defined in terms of the bias and variance as follows:

$$\text{MSE}(\hat{\rho}(\pi_e^\phi)) = \mathbb{E}[(\hat{\rho}(\pi_e^\phi) - \rho(\pi_e))^2] = \text{Var}[\hat{\rho}(\pi_e^\phi)] + (\text{Bias}[\hat{\rho}(\pi_e^\phi)])^2$$

$$\overset{(a)}{=} \text{Var}[\hat{\rho}(\pi_e^\phi)]$$

$$= \frac{1}{(mT)^2} \text{Var} \left( \sum_{i=1}^{mT} \frac{d_{\pi_e^\phi}(s_i^\phi) \pi_e^\phi(a_i|s_i^\phi)}{d_{\pi_{\mathcal{D}}^\phi}(s_i^\phi) \pi_{\mathcal{D}}^\phi(a_i|s_i^\phi)} r^\phi(s_i^\phi, a_i) \right)$$

where (a) is because $\hat{\rho}$ is an unbiased estimator as shown in Theorem 3.

Due to Assumptions 2 and 3, $\left( \sum_{i=1}^{mT} \frac{d_{\pi_e^\phi}(s_i^\phi) \pi_e^\phi(a_i|s_i^\phi)}{d_{\pi_{\mathcal{D}}^\phi}(s_i^\phi) \pi_{\mathcal{D}}^\phi(a_i|s_i^\phi)} r^\phi(s_i^\phi, a_i) \right)$ is a bounded value. Thus, as $mT \to \infty$, $\text{Var}[\hat{\rho}(\pi_e^\phi)] \to 0$. We then have $\lim_{mT \to \infty} \mathbb{E}[(\hat{\rho}(\pi_e^\phi) - \rho(\pi_e))^2] = 0$. Thus, the estimator $\hat{\rho}(\pi_e^\phi)$ is consistent in MSE. □

**Corollary 1.** *If Assumption 1 holds, then our estimator, $\hat{\rho}(\pi_e^\phi)$ as defined in Equation 3 is an asymptotically strongly consistent estimator of $\rho(\pi_e)$.*

**Proof** Theorem 2 showed that $\hat{\rho}(\pi_e^\phi)$ is consistent in terms of MSE. Then by applying Lemma 1, we have $\hat{\rho}(\pi_e^\phi)$ to be an asymptotically strongly consistent estimator of $\rho(\pi_e)$. That is, $\hat{\rho}(\pi_e^\phi) \overset{a.s.}{\to} \rho(\pi_e)$. □

### A.5 Tabular Experiments and Details with True Ratios

We conduct the following tabular experiment on the MDP pictured in Figure 4. All trajectories are 100 time-steps long.

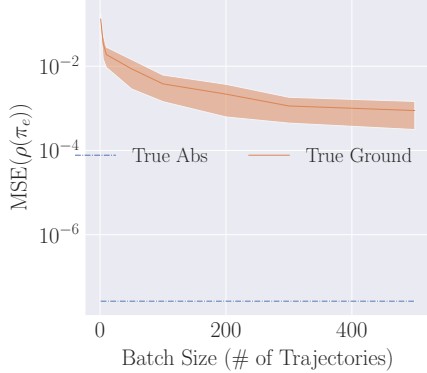

**Figure 5**: MSE vs. Batch size (# of trajectories). The vertical axis axis is log-scaled. Errors are computed over 15 trials with 95% confidence intervals. Lower is better. Since $\rho(\pi_e) = \rho(\pi_b)$ due to their symmetry, we use regular MSE instead of relative.

We conduct an experiment on the TwoPath MDP to estimate $\rho(\pi_e)$ where we apply the ground estimator given in Equation (1) and our abstract estimator given in Equation (3), assuming *both have access to their respective true ratios*. The results of this experiment are illustrated in Figure 5(a). We can observe that the abstract estimator with the true abstract ratios produces substantially more data-efficient and lower variance OPE estimates for different batch sizes compared to the ground equivalent.

## A.6 Additional Function Approximation Experiments and Details

### A.6.1 Environment Descriptions

- **Reacher** Brockman et al. (2016). A robotic arm tries to move to a goal location. Here, $s \in \mathbb{R}^{11}$ and $a \in \mathbb{R}^2$. Since the reward function is the Euclidean distance between the arm and goal, $\phi$ extracts only the arm-to-goal vector, and angular velocities from the ground state, resulting in $s^\phi \in \mathbb{R}^4$. All trajectories are 200 time-steps long.

- **Walker2D** Brockman et al. (2016). A bi-pedal robot tries to move as fast as possible. Here, $s \in \mathbb{R}^{18}$ and $a \in \mathbb{R}^6$. We use the Euclidean distance from the start location as the reward function and use a $\phi$ that extracts $x$ and $z$ coordinates and top angle of the walker's body, resulting in $s^\phi \in \mathbb{R}^3$. All trajectories are 500 time-steps long.

- **Pusher** Brockman et al. (2016). A robotic arm tries to push an object to a goal location. Here, $s \in \mathbb{R}^{23}$ and $a \in \mathbb{R}^7$. Since the reward function is the Euclidean distance between object and arm and object and goal, $\phi$ extracts only object-to-arm and object-to-goal vectors, resulting in $s^\phi \in \mathbb{R}^6$. All trajectories are 300 time-steps long.

- **AntUMaze** Fu et al. (2020). This sparse-reward task requires an ant to move from one end of the U-shaped maze to the other end. Here, $s \in \mathbb{R}^{29}$ and $a \in \mathbb{R}^8$. We use the "play" version where the goal location is fixed. Since the reward function is $+1$ only if the 2D location of the ant is at a certain Euclidean distance from the 2D goal location, $\phi$ extracts only the 2D coordinates of the ant, resulting in $s^\phi \in \mathbb{R}^2$. All trajectories are 500 time-steps long.

### A.6.2 Oracle $\rho(\pi_e)$ Values

On each domain, we executed $\pi_e$ for 200 episodes and averaged the results.

### A.6.3 Policies

For each of the domains, we used the following policies:

- Reacher: We trained a policy using PPO Schulman et al. (2017). $\pi_e$ was the trained policy after 100k time-steps with a standard deviation of $0.1$ on the action dimensions while $\pi_b$ used $0.5$ as the standard deviation.

- Walker2D: We trained a policy using PPO Schulman et al. (2017). $\pi_e$ was the trained policy after 100k time-steps with a standard deviation of $0.1$ on the action dimensions while $\pi_b$ used $0.5$ as the standard deviation.

- Pusher: We trained a policy using PPO Schulman et al. (2017). $\pi_e$ was the trained policy after 100k time-steps with a standard deviation of $0.1$ on the action dimensions while $\pi_b$ used $0.5$ as the standard deviation.

- AntUMaze: We used the policies made available Fu et al. (2021). $\pi_e$ was the final 10th snapshot saved and $\pi_b$ was the 5th snapshot. Each also had $0.1$ standard deviation on the action dimensions.

### A.6.4  Trajectory Length

For each of the domains, the trajectory length is: 200 for Reacher, 500 for Walker2D, 300 for Pusher, and 500 for AntUMaze.

### A.6.5  Hyperparameters

For BestDICE and AbstractBestDICE, we fixed the following hyperparameters:

- $\gamma = 0.995$ in all experiments.

- Neural net architecture: All neural networks are 2 layers with 64 hidden units using tanh activation.

- Unit mean constraint learning rate Zhang et al. (2020b); Yang et al. (2020): $\lambda = 1e^{-3}$.

- Optimizer: Adam optimizer with default parameters in Pytorch.

- Positivity constraint: squaring function on the last layer of the neural network.

We conducted a search for the learning rate of $\nu$ $(\alpha_\nu)$ and learning rate of $\zeta$ $(\alpha_\zeta)$, . The learning rate search for $(\alpha_\nu, \alpha_\zeta)$ was over $\{(5e^{-5}, 5e^{-5}), (1e^{-4}, 1e^{-4}), (3e^{-4}, 3e^{-4}), (7e^{-4}, 7e^{-4}), (1e^{-3}, 1e^{-3})\}$. The optimal hyperparameters $(\alpha_\nu = \alpha_\zeta)$ for each environment and batch size were:

|  | 5 | 10 | 50 | 75 | 100 | 300 | 500 | 1000 |
|---|---|---|---|---|---|---|---|---|
| Reacher | $5e^{-5}$ | $1e^{-4}$ | $1e^{-3}$ | $1e^{-4}$ | $1e^{-4}$ | $7e^{-4}$ | $1e^{-3}$ | $7e^{-4}$ |
| Walker2D | $1e^{-3}$ | $5e^{-5}$ | $5e^{-5}$ | $5e^{-5}$ | $5e^{-5}$ | $5e^{-5}$ | $5e^{-5}$ | $5e^{-5}$ |
| Pusher | $5e^{-5}$ | $5e^{-5}$ | $5e^{-5}$ | $5e^{-5}$ | $1e^{-4}$ | $5e^{-5}$ | $5e^{-5}$ | $5e^{-5}$ |
| AntUMaze | $3e^{-4}$ | $5e^{-5}$ | $5e^{-5}$ | $5e^{-5}$ | $5e^{-5}$ | $5e^{-5}$ | $5e^{-5}$ | $5e^{-5}$ |

**Table 1**: Optimal hyparameters for AbstractBestDICE on each batch size and environment.

|  | 5 | 10 | 50 | 75 | 100 | 300 | 500 | 1000 |
|---|---|---|---|---|---|---|---|---|
| Reacher | $5e^{-5}$ | $1e^{-4}$ | $5e^{-5}$ | $1e^{-4}$ | $5e^{-5}$ | $1e^{-4}$ | $3e^{-4}$ | $1e^{-3}$ |
| Walker2D | $5e^{-5}$ | $3e^{-4}$ | $7e^{-4}$ | $7e^{-4}$ | $7e^{-4}$ | $1e^{-4}$ | $3e^{-4}$ | $1e^{-4}$ |
| Pusher | $5e^{-5}$ | $1e^{-4}$ | $1e^{-4}$ | $5e^{-5}$ | $1e^{-4}$ | $5e^{-5}$ | $1e^{-4}$ | $5e^{-5}$ |
| AntUMaze | $1e^{-4}$ | $3e^{-4}$ | $5e^{-5}$ | $5e^{-5}$ | $5e^{-5}$ | $5e^{-5}$ | $5e^{-5}$ | $5e^{-5}$ |

**Table 2**: Optimal hyperparameters for BestDICE on each batch size and environment.

### A.6.6 Empirical Estimator

In practice we use a weighted importance sampling Yang et al. (2020) approach for the function approximation cases to estimate $\rho(\pi_e)$ (same for BestDICE):

$$\hat{\rho}(\pi_e^\phi) = \frac{\sum_{i=1}^N \frac{d_{\pi_e^\phi}(s_i^\phi, a_i)}{d_{\pi_\mathcal{D}^\phi}(s_i^\phi, a_i)} r^\phi(s_i^\phi, a_i)}{\sum_{i=1}^N \frac{d_{\pi_e^\phi}(s_i^\phi, a_i)}{d_{\pi_\mathcal{D}^\phi}(s_i^\phi, a_i)}}$$

### A.6.7 Misc Abstraction Details

- For Walker2D, we modified the default reward function from incremental distance covered at each time-step to distance from start location at each time-step to ensure Assumption 1 is satisfied.

- For AntUMaze, the reward function is originally $r(s')$ i.e. it is based on the *next* state that the ant moves to. To ensure Assumption 1 is satisfied, we changed this reward function to be of the *current* state, $r(s)$.

### A.6.8 Additional Results

**Additional Hyperparameter Robustness Results** In general, we can see AbstractBestDICE can be much more robust than BestDICE to hyperparameter tuning.

**Training Stability** In Figure 7 we show that $\phi$ can improve training stability.

**Abstract Quality and Data-Efficiency**. We find that not all abstractions that satisfy Assumption 1 lead to better performance. For example, the following are valid abstractions on the Reacher task: 1) the Euclidean distance between the arm and goal, $s^\phi \in \mathbb{R}$ and the 3D vector between the arm and goal, $s^\phi \in \mathbb{R}^3$ (Figure 8). However, in practice we found that these were unreliable. One possible reason for this unreliability is that these abstractions are incredibly extreme and the algorithm may be unable to differentiate between abstract state, resulting in outputting similar $\zeta^\phi(s^\phi, a) \forall s^\phi$.

## A.7 Hardware For Experiments

- Distributed cluster on HTCondor framework
- Intel(R) Xeon(R) CPU E5-2470 0 @ 2.30GHz
- RAM: 5GB
- Disk space: 4GB

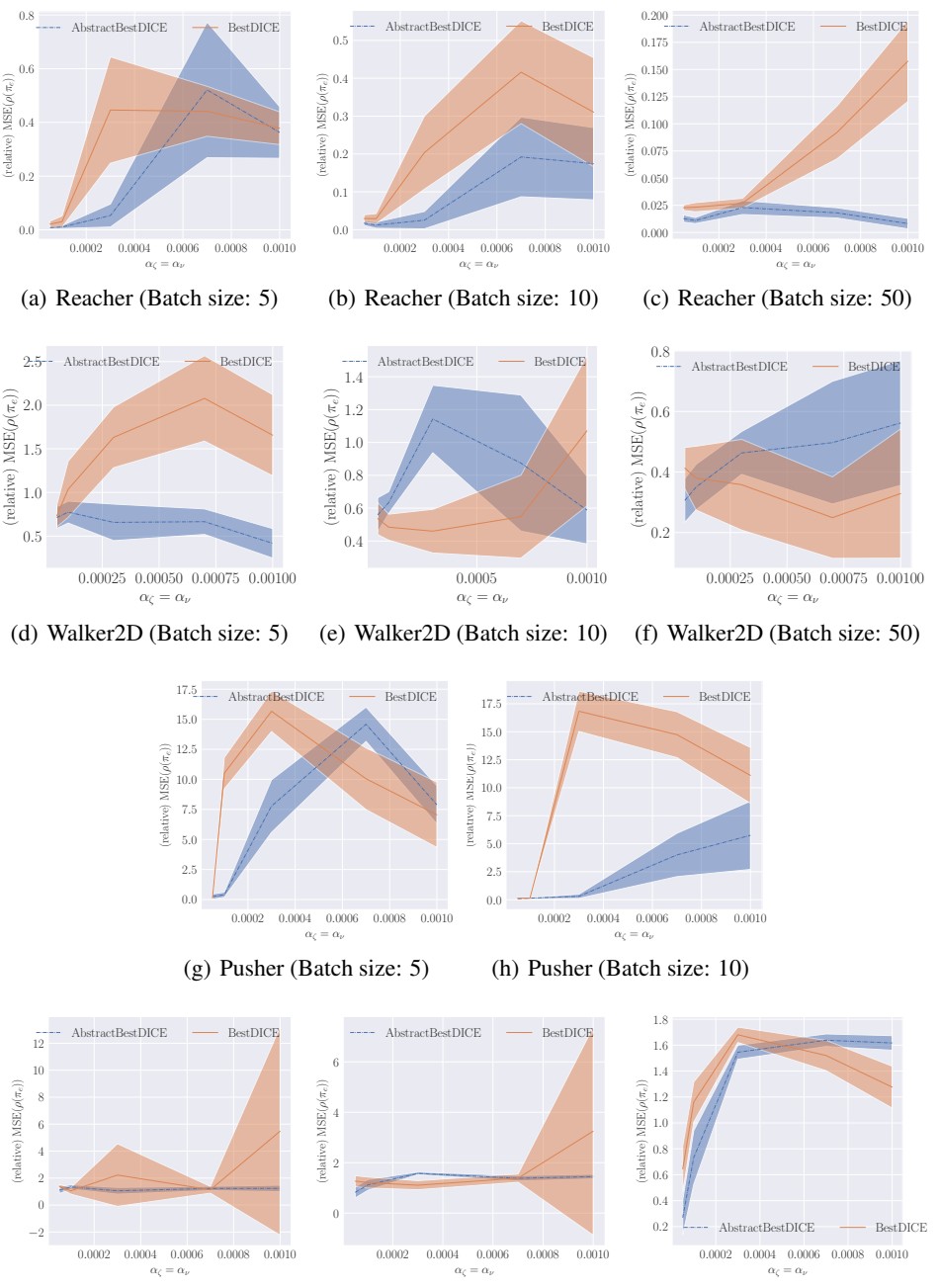

**Figure 6**: Hyperparameter sensitivity graph for BestDICE and AbstractBestDICE. $\alpha_\zeta = \alpha_\nu$ Errors are computed over 15 trials with 95% confidence intervals. Lower is better. Pusher for batch size of 50 is shown in the main paper.

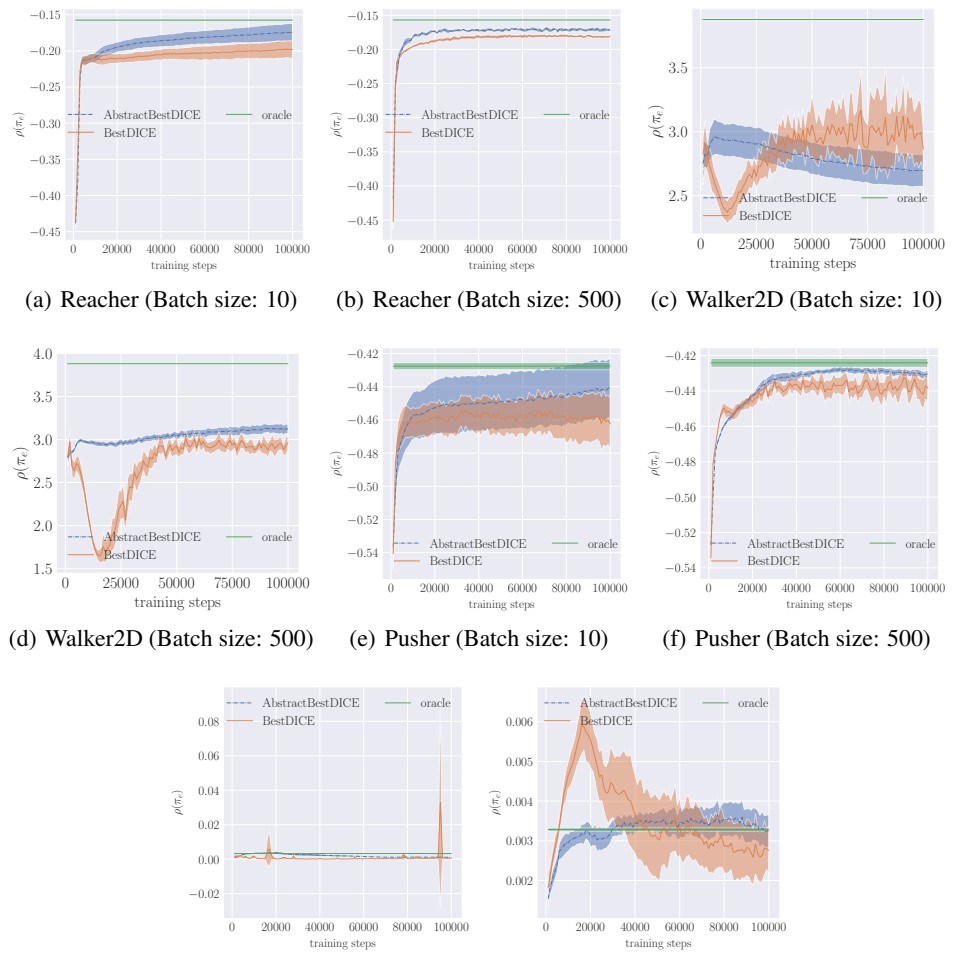

(a) Reacher (Batch size: 10) (b) Reacher (Batch size: 500) (c) Walker2D (Batch size: 10)

(d) Walker2D (Batch size: 500) (e) Pusher (Batch size: 10) (f) Pusher (Batch size: 500)

(g) AntUMaze (Batch size: 10) (h) AntUMaze (Batch size: 500)

**Figure 7**: Reward vs. Training Steps. Errors are computed over 15 trials with 95% confidence intervals. These figures illustrate the training stability of AbstractBestDICE over BestDICE. Lower is better.

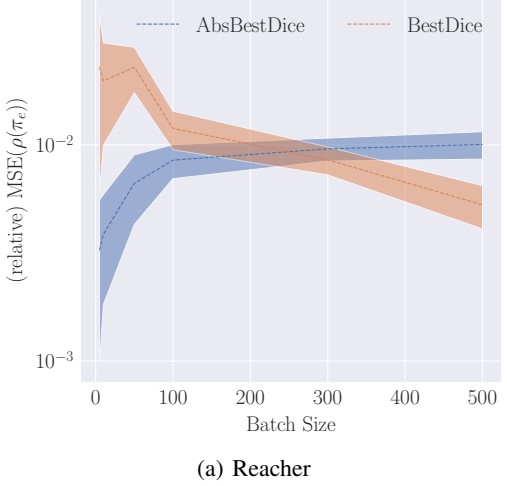

(a) Reacher

**Figure 8**: Relative MSE vs. Batch Size. $y$ axis is log-scaled. Errors are computed over 15 trials with 95% confidence intervals. This figures illustrate valid abstractions can be more data-inefficient than the ground equivalents. Lower is better.

