# OpenReview forum: "Scaling Marginalized Importance Sampling to High-Dimensional State-Spaces via State Abstraction"
_NeurIPS.cc/2022/Workshop/Offline_RL — Offline RL Workshop NeurIPS 2022_

### Official Review · Reviewer_JFit · 2022-10-19

**Rating:** 7
**Confidence:** 3

**Review:**

The authors propose doing OPE within a lower-dimensional version of the original high-D state space, with various theoretical results showing that it produces lower variance importance sampling estimates for DICE based methods.

The high level idea makes sense. My main criticism of the paper would be that the current state abstraction methods $$\phi$$ are based on domain knowledge over what parts of the state vector are relevant to reward. This feels like a lot of domain knowledge to assume for the task and the fact there are so many redundant dimensions of reward in the MuJoCo tasks is more a property of the task than anything else. (If reward is only based on these abstract states, and we know how to abstract to said states ahead of time, then why wouldn't we train RL agents in the abstracted states directly?)

I also have some questions about Assumption 1, that reward distribution from the abstracted states are identical. I would appreciate some discussion in the main text for why this is a good assumption to make about the problem, i.e. what goes wrong in either the problem or the proof if this assumption is not true. I realize the assumption is useful for the proof, and the assumption is a good condition for state abstraction in general, but it also places quite a lot of restrictions on how the method can be applied (as seen in the paper already needing to modify the Walker2D reward to fit the assumption). The paper would be stronger with either more discussion of limitations, or an argument the assumption is not as restrictive as it seems.

Once the abstraction is applied, the paper is a regular application of DICE style methods, so I feel it is important to have a stronger argument around the state abstraction - this is the new part of the paper.

---

### Official Review · Reviewer_UYWE · 2022-10-19

**Rating:** 6
**Confidence:** 4

**Review:**

This paper applies the idea of state abstraction to marginalized importance sampling (MIS) for OPE. The original idea of MIS is proposed in tabular MDP, and then it is extended to function approximation using a complicated minimax optimization with introducing lots of instability during optimization. State abstraction sounds like a reasonable way to reduce such instability. However, there are several things I am not fully convinced of. The most important one is: since there is already a state abstraction, why the DICE-style minimax optimization is still needed? I think that may be the reason for the large error bar in the empirical results. I feel the tabular style solution may be possible in this case, which could further reduce the instability as shown in this paper. At least, I strongly suggest the author add it as a baseline.